# Aerobic Exercise Alleviates Cardiac Dysfunction Correlated with Lipidomics and Mitochondrial Quality Control

**DOI:** 10.3390/antiox14060748

**Published:** 2025-06-17

**Authors:** Kunzhe Li, Sujuan Li, Hao Jia, Yinping Song, Zhixin Chen, Youhua Wang

**Affiliations:** 1Institute of Sports and Exercise Biology, School of Physical Education, Shaanxi Normal University, Xi’an 710119, China; mm1321751045@163.com (K.L.); 18768960912@163.com (S.L.); 19710125504@163.com (H.J.); syp@xafy.edu.cn (Y.S.); chenzhixin1997@snnu.edu.cn (Z.C.); 2Institute of Sports Education, Henan College of Surveying and Mapping, Zhengzhou 450000, China

**Keywords:** heart failure, aerobic exercise, lipidomics, parasympathetic nerve, mitochondrial dysfunction

## Abstract

Cardiac adaptations induced by aerobic exercise have been shown to reduce the risk of cardiovascular disease, and the autonomic nervous system is closely associated with the development of cardiovascular disease. Aerobic exercise intervention has been shown to enhance cardiac function and mitigate myocardial fibrosis and hypertrophy in heart failure mice. Further insights reveal that cardiomyocytes experiencing chronic heart failure undergo modifications in their lipidomic profile, including remodeling of multiple myocardial membrane phospholipids. Notably, there is a decrease in the total content of cardiolipin, as well as in the levels of total lysolipid CL and the CL (22:6). These alterations disrupt mitochondrial quality control processes, leading to abnormal expressions of proteins such as Drp1, MFN2, OPA1, and BNIP3, thereby resulting in a disrupted mitochondrial dynamic network. Whereas aerobic exercise ameliorated mitochondrial damage to a large extent by activating parasympathetic nerves, this beneficial effect was accomplished by modulating myocardial membrane phospholipid remodeling and restoring the mitochondrial dynamic network. In conclusion, aerobic exercise activated the parasympathetic state in mice and attenuated lipid peroxidation and oxidative stress injury, thereby maintaining mitochondrial dynamic homeostasis and improving cardiac function.

## 1. Introduction

Heart failure represents a progressive impairment of myocardial structure and function, precipitated by diverse etiologies, and ultimately serves as the terminal manifestation of all cardiovascular pathologies [1]. It is a prevalent global and public health concern, affecting an estimated 40 million individuals worldwide [2]. Despite the testing of several drugs for the treatment of heart failure in terms of clinical and pathologic interventions, there is no effective treatment for heart failure, which remains a significant cause of morbidity and mortality worldwide [1]. The precise molecular mechanisms underlying cardiac dysfunction and adverse remodeling in heart failure remain poorly understood, hindering the search for effective treatments. Consequently, there is a critical need to explore new therapeutic targets aimed at improving cardiac function in patients with heart failure. The heart is subject to autonomic nervous system (ANS) innervation, in which the vagus nerve protects the function of left ventricular cardiomyocytes via muscarinic receptors [3]. Parasympathetic activity regulates cardiac contractility, heart rate, and conduction primarily through M2 AchR. Autonomic nervous system imbalance and reduced parasympathetic activity are commonly clinical predictors of poor survival in patients with chronic heart failure [4]. An abnormal imbalance between the sympathetic and parasympathetic nervous systems in heart failure leads to overactivity of the sympathetic nervous system, resulting in suppression of cardiac parasympathetic activity, which may exacerbate heart failure by leading to deterioration of ventricular function, structural remodeling, and apoptosis of cardiomyocytes [5]. Reduced parasympathetic activity is strongly associated with increased incidence of cardiovascular disease and overall mortality. However, the exact mechanisms involved are currently not well-established and thus require further attention and investigation [6]. Additionally, the protective effect of parasympathetic activity on the heart is closely intertwined with mitochondria [7]. It was shown that exercise enhances mitochondrial quality control and reduces cardiac hypertrophy by promoting parasympathetic activity through M2 AchR expression [8]. As among the most enriched mitochondrial organs, the heart is notably hypersensitive to hypoxia-induced mitochondrial dysfunction, and prolonged hypoxia can lead to mitochondrial breaks, which are relevant to the pathogenesis of heart failure and other related diseases. Mitochondrial dynamics are controlled by the mitochondrial GTPase family, including proteins such as OPA1, Mfn2, and DRP1. DRP1 is necessary for mitochondrial division, while OPA1 and MFN2 are required for the fusion of the inner and outer mitochondrial membranes. Mitochondrial function plays a pivotal role in cardiac energy metabolism, and mitochondrial dysfunction is considered a key contributor to cardiovascular injury [9]. Cardiolipin is a signature phospholipid of mitochondria, where it plays a pivotal role in maintaining normal mitochondrial function, including membrane structure, dynamics, mtDNA biogenesis, oxidative phosphorylation, and mitophagy [10]. Abnormal cardiolipin levels and composition lead to mitochondrial dysfunction and increased release of cytochrome C, which are associated with the pathogenesis of cardiovascular disease [10]. In heart failure, abnormal cardiolipin content leads to oxidative stress, which subsequently disrupts the normal functioning of the mitochondrial dynamic network [11]. The transport of lipids from the endoplasmic reticulum to the inner mitochondrial membrane requires the involvement of mitochondrial fusion proteins, and Mfn1 and Mfn2 can form homo-or heterooligomeric complexes to aid in lipid transport [11]. Hence, lipidomics and the mitochondrial network are intricately linked, and dysfunctions in both play a collective role in the pathological processes of cardiovascular disease. Yet, the molecular mechanisms underlying pathological remodeling in cardiovascular disease remain poorly understood.

Cardiolipin (CL) is an exclusive mitochondrial phospholipid synthesized only in the inner mitochondrial membrane and plays a crucial role in maintaining mitochondrial function, including membrane structure, dynamics, oxidative phosphorylation, and mitochondrial autophagy. CL binds to the mitochondrial fission protein Drp1 and the fusion protein OPA1 through structural domains and regulates mitochondrial dynamics via variable domain structures. CL is highly susceptible to oxidation due to its proximity to the electron transport chain location where reactive oxygen species are generated. During apoptosis, externalization of oxidized CL promotes an increase in BAX proteins, and consequently, alterations in CL can lead to mitochondrial dysfunction and disease. Lipid disturbances have been found to occur in heart failure. Thus, it has been proposed that modulation of phospholipid components such as CL may attenuate lipidomic disturbances and mitochondrial dysfunction induced by heart failure. Although the underlying mechanisms have not been elucidated, the data from this trial suggest that aerobic exercise is an effective treatment for heart failure. In addition, although aerobic exercise has been reported to improve lipidomic disturbances, whether the beneficial effects of aerobic exercise to achieve improved lipidomics in cardiovascular disease are mediated by mitochondrial dynamics has not been reported yet. To this end, the present study aimed to clarify the advantages and potential mechanisms of aerobic exercise for the cardiac function and autonomic network protection in heart failure, with an emphasis on phospholipid composition and mitochondrial quality control (Figure 1).

## 2. Materials and Methods

### 2.1. Grouping of Animal Experiments and Preparation of Heart Failure Models

Eighty male 9-week-old C57BL/6 mice were randomly divided into five groups, namely, the sham-operated group (SHAM), heart failure group (HF), heart failure + aerobic exercise group (HE), heart failure + PYR intervention group (HP), and heart failure + aerobic exercise + PYR intervention group (HEP). During the experimental feeding period, the animals were kept according to animal standards and were free to drink and eat.

Preparation of the heart failure model involved sterilizing the surgical operating table and necessary instruments prior to surgery. Sterilized instruments included forceps, scissors, curved needles (27G), L-shaped needles, and surgical sutures (6–0). The small animal ventilator was checked to ensure adequate drug supply. Weighing anesthesia, tracheal intubation, chest opening, ligation, chest closure, and suturing were performed sequentially. For the sham-operated group, the ligature was threaded only without connecting the ligature, and the rest of the steps were the same. The weighing anesthesia for TAC surgery requires the mice to be weighed before surgery and the ventilator to be adjusted to the corresponding tidal volume. The weighed mice were placed into the anesthesia box, the valve of the oxygen cylinder was opened, and the anesthetic dose was administered at 3–4 graduated values, waiting for the mice to be anesthetized while observing the anesthesia status of the mice. The anesthetic dose was lowered after the chest was closed for TAC surgery. When suturing the sternal area, the ventilator outlet was pinched for 3 s to ensure negative pressure in the thoracic cavity to avoid the pneumothorax during the last suture. After the suture was completed, the anesthesia machine was turned off, and the mice were observed until they awoke.

### 2.2. Mouse Exercise Regimen, Drug Delivery Regimen

For details on the exercise training program use in this study, please refer to the research program of Takashi Sonobe et al. [8,12,13]. Exercise training was started one week after surgery in the HE and HEP groups. The exercise intensity was moderated and included the following: 5 min of warm-up, 8–9 m/min; 50 min of formal training, 12 m/min; 5 min of relaxation, 8 m/min. Six days of training were carried out per week, at 1 h per day for 10 weeks.

For details on the PYR intervention program, refer to the study program of Bernatova I et al. [14,15,16]. The parasympathetic agonist PYR (3 mg/kg/day) was used in the HP and HEP groups and administered by gavage for 10 weeks. PYR is an AChE inhibitor that inhibits the hydrolysis of acetylcholine neurotransmitter in the synaptic gap, modulates parasympathetic activity, and acts in a way like exercise.

### 2.3. Mouse Heart Ultrasonography, HRV Assay, Sampling

Ultrasound detection: The mice were anesthetized with an anesthesia machine. After the mice entered the anesthesia state, they were put into the operating table parallel to the desktop, and some areas of the mice were dehaired and disinfected. Afterwards, the ultrasound detector was turned on, an appropriate amount of ultrasound special coupling agent was squeezed on the probe, and the ultrasound probe was probed along the long axis of the heart at roughly 11 o’clock, while the ultrasound screen imaging was observed until the cardiac imaging was clear, and then the appropriate images and videos were selected for recording. After the ultrasound detection was completed, an ultrasound probe was used to analyze the relevant indexes of cardiac function in mice.

HRV test: cardiac autonomic activity and parasympathetic activity were evaluated by frequency domain analysis of R-R intervals. Mice were gas-anesthetized and placed horizontally on the operating table before the ECG test. Continuous and smooth ECGs were recorded using a needle electrode lead electrocardiogram (ECG), as well as a PowerLab chart, and HRV power spectra of the R-R intervals were obtained by the fast Fourier transform algorithm. High frequency (HF), low frequency (LF), very low frequency (VLF), and LF/HF ratios were determined using the PowerLab chart.

Sampling: The mice were decapitated and executed, following which, the chest was promptly opened to expose the internal organs. The heart was placed in saline, rinsed of blood inside and on the surface of the organs, and then drained with filter paper and weighed. Finally, the right and left ventricles of the heart were separated, placed in liquid nitrogen for rapid freezing, and then transferred to a −80 °C refrigerator for storage. Some of the hearts used for cardiac morphology were placed in 4% paraformaldehyde tubes awaiting subsequent experimental steps.

### 2.4. HE and Masson Staining

HE staining: for HE staining, 4% paraformaldehyde-fixed hearts were removed and rinsed under running water overnight and then subjected to gradient alcohol dehydration, after which, the tissue was transparent with chloroform replacement alcohol. After clearing, the hearts were embedded in wax by longitudinal or transverse wax dipping, and we waited for the wax blocks to cool. The wax blocks were sliced on a paraffin slicer to a thickness of 5 μm. Afterward, HE staining was performed, and the slices were deparaffinized in xylene; the slices were transferred to the mixture for about 5 min. Graded alcohols of 100%, 95%, 85%, and 70% were put in, and finally, the samples were transferred to the staining solution via primary water. Hematoxylin stain was stained for 5–15 min, excess hematoxylin stain was washed using primary water, and 0.5–1% hydrochloric acid alcohol (70% alcohol configuration) was used to separate the color for a few moments. Microscopic examination was performed until the nucleus and intranuclear chromatin were clear for about 10 s. Running water was used to rinse for 15–30 min, and the nucleus was blue. The samples were washed briefly with water once and placed into a 0.1–0.5% eosin staining solution for 1–5 min and exposed, in turn, to 70%, 85%, 95%, and 100% gradient alcohol dehydration. Each sample remained for 2–3 min. In this concentration of less than 95%, the alcohol eosin is easy to decolorize and is xylene transparent. After about 10 min., sealing was performed as follows: we wiped off the section around the excess of xylene, quickly added the appropriate number of drops of neutral gum, and then added the seal slides with coverslips and waited for pictures to be taken.

Masson staining: paraffin embedding, dewaxing, and rehydration treatment was carried out as described above. Hematoxylin staining of cell nuclei was performed for 5 min, followed by rinsing with tap water, 1% hydrochloric acid alcohol differentiation for several seconds, tap water rinsing again, and then rinsing with running water for several minutes until the sample returned to blue. Lichun red staining was performed for 5–10 min, followed by a rapid rinse in distilled water. The sections were treated with phosphomolybdic acid for 3–5 min, stained with aniline blue for 5 min, washed with 1% phosphomolybdic acid for 1 min, dehydrated sequentially in a gradient of alcohol, removed from xylene, dried slightly, sealed with neutral gum, and left until filming occurred.

### 2.5. Western Blotting Protein Test

Firstly, the required sample was weighed for extraction. An appropriate amount of lysate was added, and the sample was cut with scissors on ice. Subsequently, the sample was homogenized using a homogenizer, followed by centrifugation in a precooled centrifuge. After centrifugation, the supernatant was aspirated, using a pipette gun, into a new EP tube. Protein quantification was conducted using the BCA kit. After quantification, the samples were heated in a metal bath until the proteins were denatured, and then it was stored in portions in a −80 °C refrigerator. Electrophoresis buffer and a polyacrylamide gel of specific concentration were prepared. Electrophoresis buffer was added to the electrophoresis tank, and protein samples were loaded. After electrophoresis, the corresponding bands were cut out according to the marker, followed by incubation with primary and secondary antibodies. Finally, an appropriate amount of luminescent solution was applied to the bands for luminescence detection using a gel imaging system.

### 2.6. Lipidomics Assay

Lipidomics structural composition alteration assay: total lipids were extracted and separated from the experimental groups; cardiac tissues were homogenized in 2:1 (chloroform: methanol, vol/vol) by filtration to remove the mixture and cellular debris. The homogenizer and collected cellular debris were rinsed with fresh solvent mixture, and the rinse solution was mixed with the previous filtrate before adding 0.73% NaCl aqueous solution, resulting in a final solvent system of chloroform/methanol/water (vol/vol/vol) of 2:1:0.8. The lipid extracts were finally rinsed with nitrogen, capped, and stored at −20 °C. Lipid structural composition was analyzed by a triple quadrupole mass spectrometer (Thermo Electron TSQ Quantum Ultra, Trzin, Slovenia) controlled by Xcalibur (4.0, Thermo Fisher Scientific) system software. All mass spectra and tandem mass spectra were acquired automatically by a customized sequence subroutine under the Xcalibur software [10,17].

### 2.7. Data Collection and Statistical Treatment

The experimental bands of protein immunoblotting reactions were analyzed and organized using the software ImageJ(1.52a). All statistical data were subjected to statistical image preparation, as well as statistical tests using Prism 8.4 (results contain the mean + standard deviation of at least 3 independent experiments). Differences between groups were statistically analyzed by one-way ANOVA, two-way ANOVA, or independent *t*-tests. Values of *p* < 0.001 (***), *p* < 0.01 (**), and *p* < 0.05 (*).

## 3. Results

### 3.1. Aerobic Exercise Improved Cardiac Function and Following Heart Failure

With the use of mice with TAC-induced heart failure, we investigated whether aerobic exercise could attenuate heart failure-induced cardiac dysfunction. Aerobic exercise has been previously shown to improve cardiac function by modulating parasympathetic activity, and the medicine PYR inactivates cholinesterase and exerts a similar utility to exercise; therefore, we used the PYR-administered group as a positive control group. Mice in the sham surgery group served as controls, and we analyzed survival rates and pathophysiologic mechanisms associated with heart failure, including preserved ejection fraction heart failure and left ventricular dysfunction, 10 weeks after TAC surgery. The results of the cardiac function tests (Figure 2) showed that the average ejection fraction (EF) and shortening fraction (FS) in the SHAM group were 59.71% and 29.16%, respectively. Compared to the SHAM group, the LVIDd and LVIDs in the HF group were significantly increased (*p* < 0.01, *p* < 0.05), the left ventricular cavity expanded, and the average EF and FS were 34.08% and 12.83%, respectively, which decreased significantly (*p* < 0.05). EF < 40% and FS < 20% indicated severe impairment of heart function in the heart failure mice. Consistent with the study results, TAC caused heart failure and left ventricular dysfunction in mice. Compared to the HF group, all parameters in the HE, HP, and HEP groups improved significantly; the average EF and FS in the HE group were 55.05% and 24.17%, respectively (*p* < 0.01), and LVIDd and LVIDs decreased significantly (*p* < 0.01); the average EF and FS in the HP group were 59.8% and 27.05%, respectively (*p* < 0.01), and LVIDd and LVIDs decreased significantly (*p* < 0.01); the average EF and FS in the HEP group were 62.47% and 29.58%, respectively (*p* < 0.01), and LVIDd and LVIDs decreased significantly (*p* < 0.01). These data support the cardioprotective ability of aerobic exercise and pyridostigmine in heart failure, via its contribution to the upregulation of cardiac contractility.

### 3.2. Aerobic Exercise Alleviates Autonomic Nervous System Dysfunction Under Heart Failure

Consistent with the study results, heart failure leads to autonomic neuropathy and a decrease in parasympathetic activity. Heart rate variability (HRV) reflects the time between adjacent heartbeats and can reflect the regulatory role of the autonomic nervous system. The HRV analysis calculated by electrocardiogram (Figure 2) showed that LF (nu) was significantly increased (*p* < 0.01) in the HF group compared to the SHAM group, and LF (nu) was significantly reduced (*p* < 0.01) in the HE, HP, and HEP groups compared to the HF group. HF (nu) was significantly reduced (*p* < 0.01) in the HF group compared to the SHAM group, and HF (nu) was significantly increased (*p* < 0.01) in the HE, HP, and HEP groups compared to the HF group. As expected, aerobic exercise and pyridostigmine intervention significantly improved autonomic neuropathy in heart failure and inhibited the decrease in parasympathetic activity.

### 3.3. Aerobic Exercise Reverses Myocardial Fibrosis and Cardiomyocyte Hypertrophy with Heart Failure

In addition to left ventricular systolic dysfunction, heart failure can lead to left ventricular hypertrophy and fiberization. Masson staining results (Figure 3) showed normal myocardial cell morphology in the SHAM group, with almost no collagen fibers between cells and normal vascular tissue. In contrast, cardiomyocytes in the HF group exhibited disorganization, extensive blue collagen fiber deposition, and pronounced fibrosis, particularly near blood vessels. Statistical results indicated that TAC-induced heart failure led to severe myocardial fibrosis and extensive myocardial cell hypertrophy. Compared to the HF group, the heart and vascular fibrosis and pathological tissue morphology in the HE, HP, and HEP groups improved significantly. The myocardial cell cross-sectional area was measured using HE and WGA staining. These results indicated that aerobic exercise and pyridostigmine intervention significantly attenuated miocardial fibrosis and myocardial hypertrophy and improved pathological remodeling of the myocardium in heart failure.

### 3.4. Aerobic Exercise Regulates Lipidomic Abnormalities in Heart Failure

As mentioned previously, mitochondrial membrane lipids (e.g., CL, PC, PA, PE, and SM) are critical for oxidative phosphorylation and the maintenance of mitochondrial function, with CL being essential for the stability of the mitochondrial respiratory chain complex. CL is mainly produced by PG, but PA can be converted to PG via Gep4 and is involved in the production of CL, which is closely related to the function of mitochondrial splitter proteins DRP1; the functions of the fusion protein OPA1 are closely related to this (Figure 4G). In addition, CL peroxides further damage to neighboring acyl chains, triggering a vicious cycle known as CL peroxidation, oxidative stress, and mitochondrial dysfunction. Therefore, we focused on CL and PG as indicators of focus. SM can be synthesized from PC and Cer via SMS1 and has a role in alleviating neurological dysfunction. It has been shown that PC deficiency leads to TNFα-induced NFκB activation and upregulation of inflammatory factor expression. In addition, PE is also converted to PC via PEMT, which is involved in SM generation (Figure 4K). Lipidomics results (Figure 4A–H) showed that total CL was significantly lower (*p* < 0.05), and PG, PE, and PC were decreased in the HF group compared to the SHAM group; CL, PE, and PC were elevated in the HE, HP, and HEP groups compared to the HF group; PG was elevated in the HE and HEP groups compared to the HF group and was significantly elevated in the HP group (*p* < 0.05). Compared to the SHAM group, total 4-HNE, LPE, and PA were elevated in the HF group; 4-HNE and PA were decreased in the HE, HP, and HEP groups compared to HF; LPE was decreased in the HE and HEP groups and significantly decreased in the HP group compared to the HF group (*p* < 0.01). Total SM was decreased in the HF group compared to the SHAM group, and in the HE, HP, and HEP groups, total SM was elevated compared to the HF group. Our results show that exercise alleviates lipidomic disorders and reduces lipid peroxidation and mitochondrial damage in heart failure.

CL, PE, and PC in mitochondria are all essential for OXPHOS, and these lipids directly interact with and affect the activity of mitochondrial complexes I–V. We measured PE content in the left ventricle and found that PE content was reduced in the hearts of heart failure mice, while PE content was differentially elevated in the hearts of intervention group mice (Figure 4 and Figure 5). Abnormal expression of phosphatidylethanolamine (PE) and a tendency toward elevated levels of the lipid peroxidation product 4-HNE further indicate that mitochondrial dysfunction and lipid damage form a vicious cycle in heart failure.

More detailed information on the molecular structure of phospholipids obtained by mass spectrometry (Figure 4G,K) showed that TLCL and CL (22:6) were significantly reduced in the HF group compared to the SHAM group (*p* < 0.01, *p* < 0.001) and that TLCL and CL (22:6) were elevated in the HE, HP, and HEP groups compared to HF group. Figure 4L,M shows that PG (16:0–18:1) and PC (D18:0–22:6) were significantly reduced in the HF group compared to the SHAM group (*p* < 0.001, *p* < 0.01), and PG (16:0–18:1) and PC (D18:0–22:6) were elevated (*p* < 0.05, *p* < 0.01, *p* < 0.001) in the HE, HP, and HEP groups compared to the HF group. The four fatty acyl chains of cardiolipin are dominated by linoleic acid (C18:2) in the heart and other metabolic tissues. This unique acyl composition plays a key role in supporting heart health.

### 3.5. Aerobic Exercise Modulates Autonomic Nervous System to Reduce Mitochondrial Dysfunction and Subsequent Inflammation

Acetylcholine primarily mediates the protective effects of the parasympathetic nervous system through the M2 subtype of the muscarinic acetylcholine receptor in cardiac cells. We aimed to determine whether aerobic exercise can achieve restoration of mitochondrial function by regulating the expression of these proteins. Western blotting results (Figure 6) showed that M2 protein expression was significantly elevated in the HF group compared with the SHAM group (*p* < 0.05); M2 protein expression was decreased in the HE and HP groups and significantly decreased in the HEP group compared with the HF group (*p* < 0.05). Compared with the SHAM group, Mfn2 protein expression was significantly upregulated in the HF group (*p* < 0.01); compared with the HF group, Mfn2 protein expression was significantly downregulated in the HE and HP groups, as well as in the HEP group (*p* < 0.01). OPA1 protein expression was significantly downregulated in the HF group compared with the SHAM group (*p* < 0.01); OPA1 protein expression was significantly upregulated in the HE group, as well as in the HEP group, compared with the HF group (*p* < 0.05), and OPA1 protein expression was upregulated in the HP group. DRP1 protein expression was significantly downregulated in the HF group compared with the SHAM group (*p* < 0.01); DRP1 protein expression was significantly upregulated in the HE group, as well as in the HEP group, compared with the HF group (*p* < 0.01, *p* < 0.05), and DRP1 protein expression was upregulated in the HP group. BNIP3 protein expression was downregulated in the HF group compared to the SHAM group; it was significantly upregulated in the HP group, as well as in the HEP group, compared to the HF group (*p* < 0.05). The results indicate that the expression of endomembrane fusion proteins is decreased in the myocardium of heart failure mice, while the expression of outer membrane fusion proteins is increased, suggesting mitochondrial dynamic dysfunction. Aerobic exercise and pyridostigmine intervention may stabilize the balance of mitochondrial dynamics in heart failure, indicating that aerobic exercise could activate the parasympathetic nervous system to rectify abnormal mitochondrial fission and fusion in pathological conditions.

Moreover, aerobic exercise and pyridostigmine interventions decrease myocardial inflammation and oxidative stress resulting from mitochondrial dysfunction. Western blotting results (Figure 6F–H) showed that TXNIP protein expression was significantly upregulated in the HF group compared to the SHAM group (*p* < 0.05) and significantly downregulated in the HE, HP, and HEP groups compared to the HF group (*p* < 0.01, *p* < 0.05, *p* < 0.001). ASC protein expression was upregulated in the HF group compared to the SHAM group and downregulated in the HE, HP, and HEP groups compared to the HF group. BAX protein expression was significantly upregulated in the HF group compared to the SHAM group (*p* < 0.05) and downregulated in the HE and HEP groups compared to the HF group. The results indicated that under pathological conditions, the levels of inflammation and apoptosis in myocardial tissue were high. Aerobic exercise and pyridostigmine intervention could reduce inflammation and apoptosis in heart failure, diminishing the area of myocardial fibrosis in mice with heart failure.

## 4. Discussion

Aerobic exercise has been reported to ameliorate mitochondrial dysfunction and autonomic nervous system disorders in heart failure; however, its role in mediating lipidomics has not yet been elucidated [18,19]. The current study shows that aerobic exercise improves lipidomic disturbances and restores mitochondrial quality control. Additionally, parasympathetic nerves modulate phospholipid components involved in mitochondrial dynamics via M2 receptors. As a signature phospholipid of mitochondria, cardiolipin plays a crucial role in maintaining mitochondrial functions such as membrane structure, dynamics, oxidative phosphorylation, and mitochondrial autophagy. Cardiolipin accounts for almost 10% of the total lipid content of the organelle, which is associated with protein binding of the OXPHOS complexes I through V and maintenance of stability of overloaded species, and thus plays a vital role in participating in oxidative phosphorylation and maintaining mitochondrial function [20]. CL plays a role as the “glue” of the mitochondrial respiratory complex and is considered essential for the proper function of the electron transport chain. In addition, CL is located near the ROS-generating site in the inner mitochondrial membrane, and lipid peroxides generated by oxidation are trapped in the mitochondria, making them more toxic than other forms of ROS. CL peroxidation disrupts the binding of CL to cytochrome C and affects the activity of complexes I, III, and IV of the mitochondrial respiratory chain, while the ROS burst causes a massive loss of CL leading to mitochondrial damage [21]. These demonstrate the critical role of CL peroxidation in mitochondrial dysfunction. Research has shown that reduced cardiolipin levels in the heart cause dilated cardiomyopathy, but the cause of cardiolipin depletion in patients with heart failure is unknown [22,23]. In parallel, we also examined the lipid composition of the right heart, and similar to the results of the left heart, these showed a decrease in CL (Appendix A).

Similarly, our findings suggest that heart failure causes cardiolipin depletion, mitochondrial dysfunction, and oxidative stress, leading to myocardial inflammation, fibrosis, and apoptosis. In contrast, these pathologic defects can be attenuated by aerobic exercise and pyridostigmine pharmacologic interventions. The present study provides detailed additional information on the beneficial effects of aerobic exercise in chronic heart failure, whereby aerobic exercise significantly improves cardiac function in heart failure, reverses pathological hypertrophy, reduces fibre deposition, and restores mitochondrial and lipid histological disturbances. The potential mechanism may be the endogenous modulation of the autonomic nervous system by aerobic exercise, thereby reducing the aberrant binding of phospholipid components to mitochondria-associated proteins. Furthermore, aerobic exercise decreases myocardial fibrosis, hypertrophy, and contractile dysfunction by inhibiting inflammatory factor expression and oxidative stress and improving cardiomyocyte remodeling. Overall, aerobic exercise exerts an effective and comprehensive protective effect against chronic heart failure. Additionally, we hypothesize that aerobic exercise effectively attenuates various dysfunctions induced by heart failure by restoring mitochondrial function mainly through cardiolipin [10,11]. Meanwhile, PA and PG can participate in mitochondrial function as substrates for cardiolipin. PA formation occurs on the outer surface of the mitochondrial outer membrane and endoplasmic reticulum and can be transferred to the inner mitochondrial membrane through the membrane gap and the outer mitochondrial membrane. PG is formed through Tam41, Pgs1, Gep4, and Crd1 and is finally converted to CL at the inner mitochondrial membrane by Taz1 [20]. In support of aerobic exercise modulating CL to ameliorate mitochondrial dysfunction in cardiovascular disease, we further demonstrated that aerobic exercise increases CL (18:2) and PG, which increase the synthesis of substrates for CL, an improvement in left ventricular systolic function, as well as a reduction in myocardial fibrosis in heart failure. Cardiolipin is mostly located in the inner mitochondrial membrane and can bind to part of the structural domain of the inner mitochondrial membrane fusion protein OPA1; together, they coordinate roles in mitochondrial fusion [24,25]. Our data show that the inner membrane fusion proteins OPA1 and CL, which promote mitochondrial fusion, are simultaneously reduced in the cardiac myocardium of mice with heart failure, resulting in insufficient protein bridge connections between mitochondrial inner membranes and indirectly worsening mitochondrial fragmentation. In addition, the Drp1 B insert or a variable structural domain of the mitochondrial fragmentation protein Drp1 is important for mitochondrial recruitment and autophagy, among other processes. The Drp1 B insert is dispensable for mitochondrial recruitment, association with Mff, and basal and protonophore-stimulated mitochondrial fission. Cardiolipin can interact with the four lysine modules of the B insert domain to regulate Drp1 function [23]. Consistent with this notion, our experimental results suggest that aerobic exercise upregulates cardiolipin, modulates the expression of mitochondrial dynamics-related proteins OPA1 and Drp1, promotes mitochondrial fusion, reduces mitochondrial fission, and finally, supports CL in the mitochondrial etiology of HF-induced myocardial dysfunction.

Additionally, diminished parasympathetic activity is an important cause of mitochondrial dysfunction in heart failure [26]. It has been observed that increasing parasympathetic nerve activity through vagus nerve stimulation promotes upregulation of the mitochondrial fusion protein OPA1, thereby improving imbalance in the mitochondrial dynamic network and reducing oxidative stress and myocardial injury. These cardioprotective effects on the heart are correlated with those of muscarinic acetylcholine receptors (mAChR) [27]. Similarly, the current study found that the combined intervention of aerobic and pharmacologic exercise had the same cardioprotective effects in treated heart failure mice. Aerobic exercise restored M2 protein expression to a level similar to that of normal controls, reduced LF and LF/HF ratio, alleviated heart failure-induced autonomic nervous system disorders, and thus restored the expression of mitochondrial division, fusion, and autophagy-related proteins. Since the mode of mitochondrial energy supply in heart failure is changed from fatty acid oxidation to glucose metabolism, it leads to impaired energy metabolism, and various therapeutic strategies of fatty acid oxidation stimulation currently show great advantages in the prevention and treatment of heart failure, whereas phosphatidylethanolamine (PE), phosphatidylcholine (PC), phosphatidylserine (PS), and sphingomyelin (SM) are precisely the important constituent groups of phospholipids [28,29]. Phospholipids are highly correlated with neurological function, and SM is abundantly present in the brain and neural tissues and can be synthesized by PC and ceramide Cer via the enzyme SM synthase 1 (SMSA), which has the ability to promote synaptic transmission and ameliorate neurological dysfunction. A defective PC layer contributes to inflammation, while PC can inhibit TNFα-induced NFκB activation, downregulate the expression of proinflammatory genes, and increase ATP-dependent ubiquitin proteasome formation, along with attenuating neutrophil-mediated inflammatory responses [30]. From the lipidomics experiments, it was observed that the contents of PC and SM in the hearts of heart failure mice decreased. Additionally, phosphatidylethanolamine, a precursor for PC synthesis, also showed a decreasing trend. These findings suggest a close relationship between cardiac injury in heart failure mice and neurological dysfunction. Excessive accumulation of ROS causes multiple losses to the cells while decreasing GSH content in the cells. GSH, a water-soluble tripeptide composed of the amino acids glutamine, cysteine, and glycine, exists in the cellular milieu in either its oxidized state, GSSG, or its reduced state, GSH [31]. Functionally, glutathione plays an influential role as an antioxidant enzyme in the elimination of reactive oxygen species and lipid peroxidation. TXNIP promotes the reduction of oxidized cysteine residues on cellular proteins, protects cells from oxidative stress, and works critically in regulating oxidative stress kinetics [32]. In this work, we also found that increased expression of lipid peroxidation 4-HNE in the hearts of heart failure mice prompted TXNIP to protect the damaged heart through compensatory elevation. Mechanistically, TXNIP promotes GSH effects in response to lipid peroxidation in heart failure. However, TXNIP overexpression promotes an increase in the inflammatory activator NLRP3 in the heart, reduces cardiomyocyte survival, and causes cardiac scar formation, which ultimately leads to cardiac dysfunction [33,34,35]. CL is able to directly bind pro-casp-11, while exposure of CL on the outer mitochondrial membrane leads to enhanced association of Casp-1 and NLRP3 with mitochondria, recruiting more NLRP3 and inducing activation of inflammatory vesicles [36]. In contrast, the improvement in neurological function, increase in myocardial PC content, and decrease in 4-HNE content after aerobic exercise intervention suggest that exercise reduces inflammation, lipid peroxidation, and oxidative stress injury in the heart of heart failure mice through multiple mechanisms. The current evidence all supports the alteration of phospholipid components, such as CL and 4-HNE, by exercise in heart failure, as well as the associated improvement in mitochondrial function. Therefore, we hypothesized that aerobic exercise may improve cardiac function via modifying the ANS, which modulates mitochondrial dynamics and related phospholipid components.

The American College of Cardiology/American Heart Association Joint Committee states that the safety and efficacy of exercise training in patients with heart failure has been demonstrated, and that aerobic exercise can be a key component of secondary prevention guidelines for patients with chronic heart failure (New York Heart Association, NYHA Class I–III). Aerobic exercise can be a key component of the secondary prevention guidelines for patients with chronic heart failure (NYHA Class I–III), but exercise programs should follow ESC and AHA guidelines. Low-intensity exercise (<40% VO_2_ peak) is recommended for patients with NYHA Class III for the first 1–2 weeks, gradually increasing to 50–70% VO_2_ peak as appropriate, and ultimately to a target dose of 85% VO_2_ peak. Thus, the clinical prescription of exercise for patients with heart failure needs to be combined with NYHA classification and exercise testing for more detailed judgment [1].

## 5. Conclusions

In conclusion, aerobic exercise can modulate parasympathetic activity in heart failure mice which, in turn, improves mitochondrial dysfunction and lipidomics. However, clinical exercise prescription for heart failure patients still requires relevant exercise experiments and grading before further implementation.

## Figures and Tables

**Figure 1 antioxidants-14-00748-f001:**
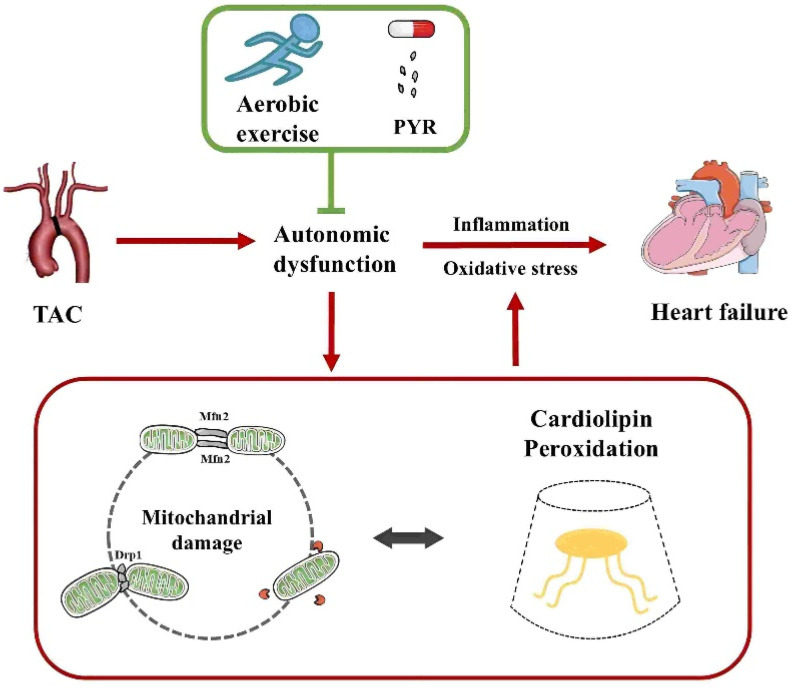
Aerobic exercise ameliorates heart failure by regulating mitochondrial damage and cardiolipin peroxidation through autonomic dysfunction. TAC, transverse aortic constriction; PYR, pyridostigmine.

**Figure 2 antioxidants-14-00748-f002:**
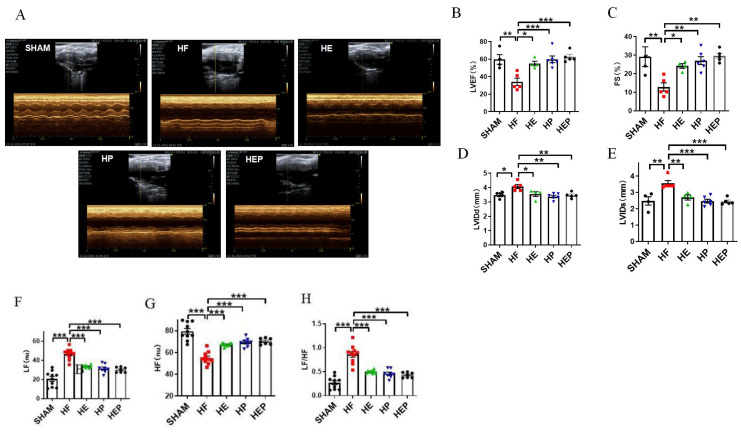
Effects of aerobic exercise on left ventricular systolic and diastolic function and on autonomic system and parasympathetic function after TAC. (**A**–**E**) LVEF, FS, LVIDd, and LVIDs were calculated by M-mode echocardiography in mice after TAC. (**A**–**H**) LF, HF, and LF/HF were collected by HRV. Data are expressed as mean ± SD; LVEF, left ventricular ejection fraction; FS, left ventricular fraction shortening; LVIDd, left ventricular internal dimension systole; LVIDs, left ventricular internal dimension in systole; HRV, heart rate variability; LF, low frequency; LF (nu), low-frequency power; HF, high frequency; HF (nu), high-frequency power. *, *p* < 0.05; **, *p* < 0.01; ***, *p* < 0.001.

**Figure 3 antioxidants-14-00748-f003:**
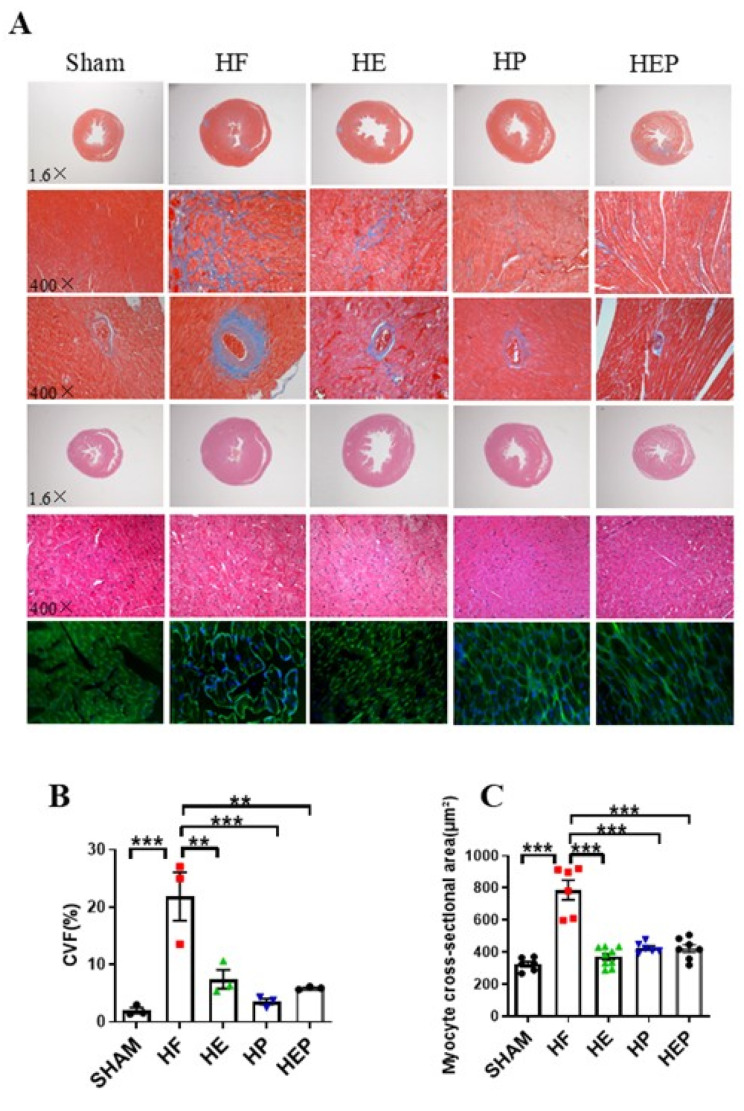
Aerobic exercise reduced TAC-induced cardiomyocyte hypertrophy and fibrosis. (**A**) Representative heart sections were stained with HE, Masson’s stain, and WGA to assess cardiomyocyte hypertrophy and fibrosis, respectively. Red area represents normal heart tissues; blue area represents myocardial fibrosis. (**B**) CVF%. (**C**) Myocyte cross-sectional area, calculated from WGA statistics. CVF, collagen volume fraction. Data are expressed as mean ± SD. **, *p* < 0.01; ***, *p* < 0.001.

**Figure 4 antioxidants-14-00748-f004:**
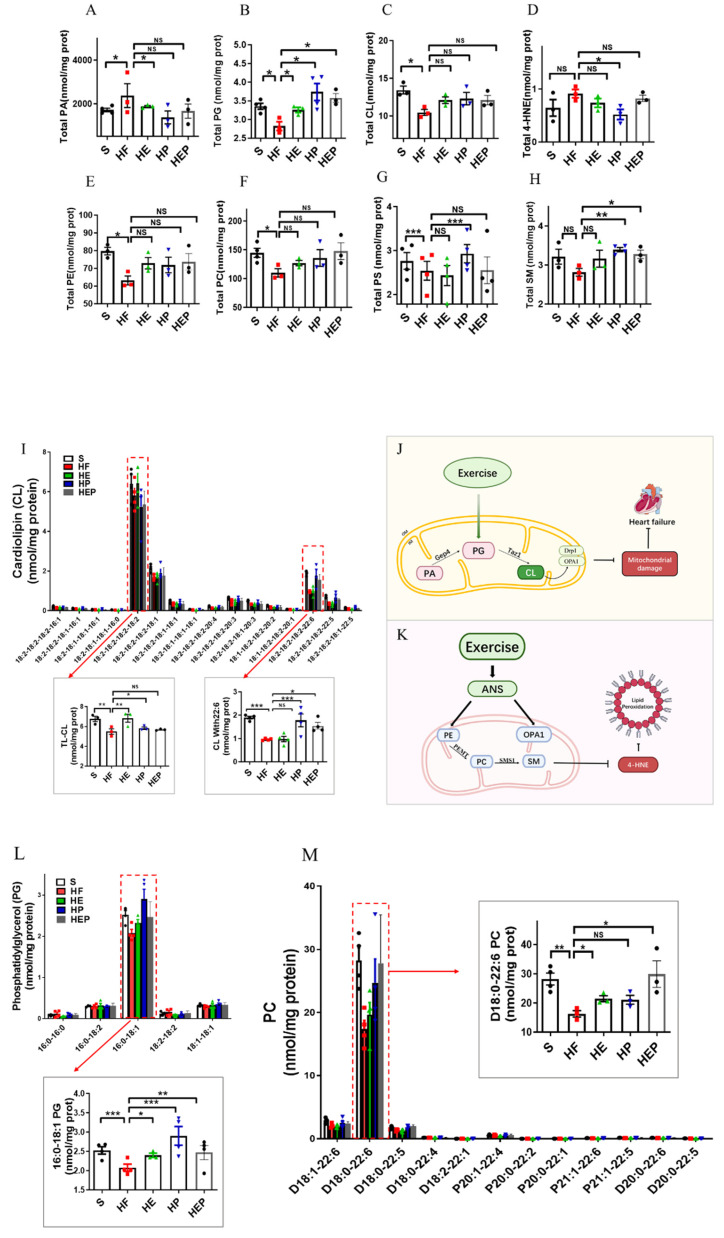
Analysis of CL acyl composition in the heart by electrospray ionization (ESI)/MS. Aerobic exercise restores lipidomic disturbances and cardiolipin acyl composition in left heart cardiomyocytes of heart failure mice. (**A**–**H**) The levels of total PA, PG, CL, 4-HNE, PE, PC, PS, and SM. (**I**–**K**) Exercise restored the levels of TLCL and CL (22:6). (**L**,**M**) Exercise restored the PG level concurrently with increased PG (16:0–18:1) and PC (D18:0–22:6). Exercise modulates mitochondria-associated dynamics-related proteins through lipidomics to inhibit lipid peroxidation and ameliorate heart failure. (**I**,**L**,**M**) used the same groups of sham, heart failure, HE, HP, and HEP; n = 3–5 per group. Data are expressed as mean ± SD. *, *p* < 0.05; **, *p* < 0.01; ***, *p* < 0.001, NS means not significant.

**Figure 5 antioxidants-14-00748-f005:**
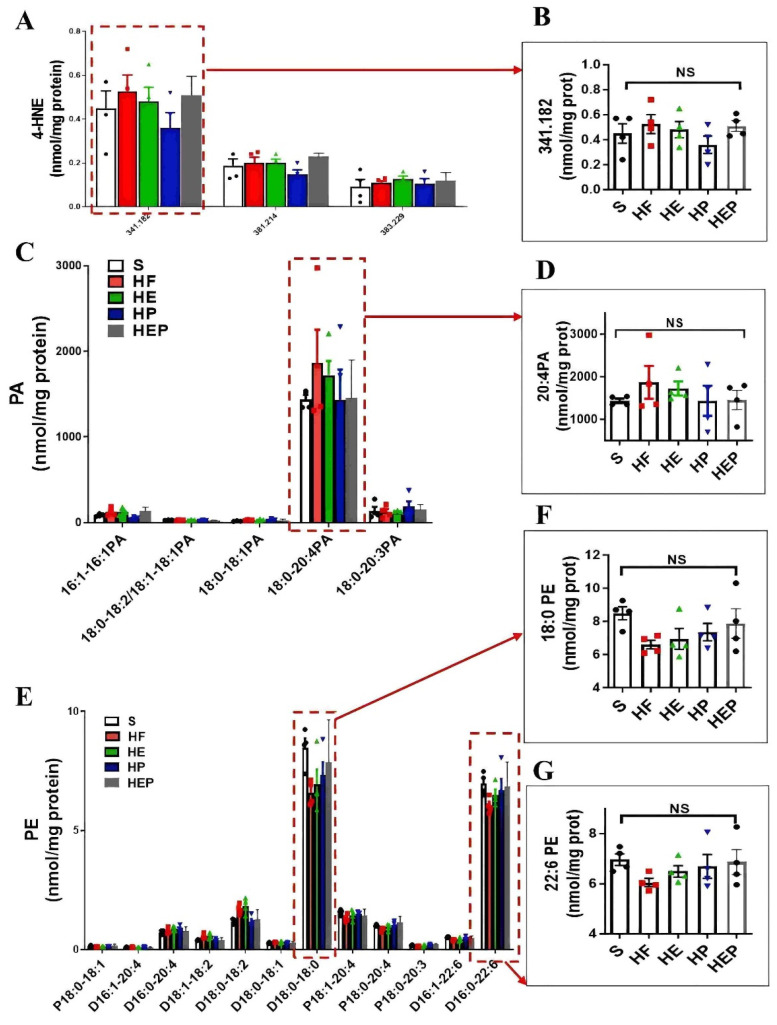
Aerobic exercise can regulate lipidomic disturbances and cardiolipin acyl composition in left heart cardiomyocytes of heart failure mice. (**A**,**B**) Exercise can regulate the levels of 4-HNE and 4-HNE (341.82) and the levels of PA and PA (20:4PA) (**C**,**D**). (**E**–**G**) Aerobic exercise regulated the cardiac PE and PE (18:0 PE) and PE (22:6 PE) level. (**A**,**C**,**E**) used the same groups of sham; heart failure; and HE, HP, and HEP groups; n = 3–5 per group. Data are expressed as mean ± SD. NS means not significant.

**Figure 6 antioxidants-14-00748-f006:**
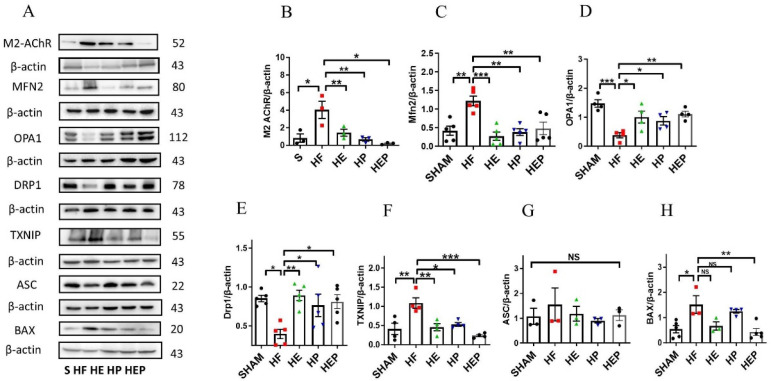
Effects of 10-week aerobic exercise on TAC-induced M2AChR, inflammation, oxidative stress, and mitochondrial quality-control-related proteins. (**A**) Western blot analysis of M2AChR, Mfn2, OPA1, Drp1, TXNIP, ASC, and BAX. (**B**–**H**) Western blot analyses of global proteins M2AChR, Mfn2, OPA1, Drp1, TXNIP, ASC, and BAX levels in mouse hearts. Data are expressed as mean ± SD. M2AChR, muscarinic acetylcholine receptor M2; Mfn2, mitofusin2; OPA1, optic atrophy 1; Drp1, mitochondrial dynamin-related protein 1; TXNIP, thioredoxin-interacting protein; ASC, recombinant Asc-type amino acid transporter 1; BAX, BCL2-associated X protein. Data are expressed as mean ± SD. *, *p* < 0.05; **, *p* < 0.01; ***, *p* < 0.001, NS means not significant.

## Data Availability

The original contributions presented in this study are included in the article. Further inquiries can be directed to the corresponding author.

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
