# Peer review of "Aerobic Exercise Alleviates Cardiac Dysfunction Correlated with Lipidomics and Mitochondrial Quality Control"

_antioxidants, 2025, doi:10.3390/antiox14060748_

Round 1

Reviewer 1 Report

Authors analyzed the potential role of 10 weeks-aerobic exercise on a mice model of TAC and CHF. The work suggests an interesting area of research and potential therapeutic strategies for CHF. Also, propose a candidate mechanism of action in these responses. The autonomic nerve stimulation and the mitochondrial function may be involved.

Firstly, the work cannot be understood without deciphering the acronyms at the first time. Also, some methodological details must be included in results before showing data.

In addition, more evidence is needed to show failure in autonomic stimulation and overall, to demonstrate connection between autonomic disruption and mitochondrial dysfunction.

In Discussion, what is the connection of lipid alterations and inflammation? And what is the role of metabolism on structural lipids (from mitochondria)?

Lipidomics has been done in the whole tissue? If so, why are you claiming on mitochondria? A mitochondrial lipidomic should have been more appropriated.

Cardiolipin is included in the lipidomic analysis. Why is being mentioned as another analyte?

A limitation of the study section is compulsory in this area. Other factors can influence the mitochondrial quality. Respiration, oxidation, mitophagy, biogenesis and metabolism among them

The PYR agonist has not been introduced. What is the aim of that? its use in rodents?

The aerobic exercise should be more documented in Methods

The section 2.1 should be rewritten.  

The beginning of 2.4 is redundant with Introduction. Why are discussion paragraph in the Results section?

The beginning of 2.5 should be a part of Introduction

In the figure 1, the cardiolipin peroxidation scheme cannot be understood.

Author Response

Dear Editors and reviewers:

Reviewer 2 Report

  • How is aerobic exercise clinically relevant in Heart failure (HF)? Many HF patients are exercise intolerant. As such, improvement observed in cardiac remodeling, structure and function might not be applicable or relevant to many patients. The authors need to comment about that and also mention it in their discussion as a potential caveat.

  • The conclusions drawn in Lines 211-227 that mitochondrial dynamics in HF is disrupted is far-fetched without any morphological data to support it. Decrease in OPA1 (fusion, cristae morphology regulator), increase in MFN2 (fusion) and decrease in Drp1 (fission) protein content alone is not sufficient to draw that conclusion which forms a major part of the mechanism that the authors mention.  Further, in lines 295-298 the authors mention that HF indirectly exacerbates mitochondrial fragmentation (more fission then?) though decrease in OPA1 is not sufficient to come to this conclusion. As such, the authors need to measure fission/fusion and look at cristae morphology as well.

  • There is discrepancy in how the authors present their conclusion regarding mitochondrial dynamics as well. In lines 224-227, they mention that, “mitochondrial division in the myocardium of heart failure mice was downregulated” meaning less fission. Further, in their discussion in lines 295-296 they mention, “heart failure-induced myocardial dysfunction indirectly exacerbates mitochondrial fragmentation” meaning more fission. The authors need to explain this apparent contradiction.

  • In Figure 5, the authors figure heading reads, “Aerobic exercise restores lipidomic disturbances and cardiolipin acyl composition in left heart cardiomyocytes of heart failure mice”. However, none of the presented results show any sort of significance in the data to warrant this strong conclusion. The authors need to revisit this finding to make sure that the extrapolated results support their conclusion.

  • What is Pyridostigmine? What is the rational behind using it in the study and in their HF animal model? The authors need to mention this as part of their introduction rather than introducing it pretty late in the discussion.

  • Please mention what kind of statistical measurement were used to identify significance between groups in each figure legend as well. For example, what kind of stats were used in Fig 4G to see the observed strong significance between S and HF (<0.001), HF and HP for an n=4? Also, please explain figure 4m-n in the text as well and elaborate a bit about the mechanism behind PG, CL, lipid peroxidation in the result section so that it is easier for reader to follow and maintain the flow of the manuscript.

  • For section 2.4, the authors need to define a little bit about the relevance of SM, PS, PE, PC and the rationale behind assessing it should also be in the results before the discussion (322-324).

In this manuscript Li et al. look at the role played by aerobic exercise and its importance in improving cardiac function, decreasing fibrosis and hypertrophy in heart failure. They show that heart failure result in decreased CL and lipidomic profile modifications disrupting mitochondrial quality process and further showed that aerobic exercise activated parasympathetic state, decreased lipid peroxidation, oxidative stress and maintained mitochondrial homeostasis and improved cardiac function. However, the authors come into strong conclusions based upon not so strong findings. As such, there are some concerns that needs to be addressed to bolster this manuscript. Please see my comments below:

Major Review points:

  • How is aerobic exercise clinically relevant in Heart failure (HF)? Many HF patients are exercise intolerant. As such, improvement observed in cardiac remodeling, structure and function might not be applicable or relevant to many patients. The authors need to comment about that and also mention it in their discussion as a potential caveat.

  • The conclusions drawn in Lines 211-227 that mitochondrial dynamics in HF is disrupted is far-fetched without any morphological data to support it. Decrease in OPA1 (fusion, cristae morphology regulator), increase in MFN2 (fusion) and decrease in Drp1 (fission) protein content alone is not sufficient to draw that conclusion which forms a major part of the mechanism that the authors mention.  Further, in lines 295-298 the authors mention that HF indirectly exacerbates mitochondrial fragmentation (more fission then?) though decrease in OPA1 is not sufficient to come to this conclusion. As such, the authors need to measure fission/fusion and look at cristae morphology as well.

  • There is discrepancy in how the authors present their conclusion regarding mitochondrial dynamics as well. In lines 224-227, they mention that, “mitochondrial division in the myocardium of heart failure mice was downregulated” meaning less fission. Further, in their discussion in lines 295-296 they mention, “heart failure-induced myocardial dysfunction indirectly exacerbates mitochondrial fragmentation” meaning more fission. The authors need to explain this apparent contradiction.

  • In Figure 5, the authors figure heading reads, “Aerobic exercise restores lipidomic disturbances and cardiolipin acyl composition in left heart cardiomyocytes of heart failure mice”. However, none of the presented results show any sort of significance in the data to warrant this strong conclusion. The authors need to revisit this finding to make sure that the extrapolated results support their conclusion.

  • What is Pyridostigmine? What is the rational behind using it in the study and in their HF animal model? The authors need to mention this as part of their introduction rather than introducing it pretty late in the discussion.

  • Please mention what kind of statistical measurement were used to identify significance between groups in each figure legend as well. For example, what kind of stats were used in Fig 4G to see the observed strong significance between S and HF (<0.001), HF and HP for an n=4? Also, please explain figure 4m-n in the text as well and elaborate a bit about the mechanism behind PG, CL, lipid peroxidation in the result section so that it is easier for reader to follow and maintain the flow of the manuscript.

  • For section 2.4, the authors need to define a little bit about the relevance of SM, PS, PE, PC and the rationale behind assessing it should also be in the results before the discussion (322-324).

Some minor revisions are:

  • Line 56- first should be mitochondria and not mitochondrial
  • Fig 2 legend random B and also the authors do not have any info about F-H
  • Fig 3: Reorder HE, Masson and WGA so that respectively makes sense. Right now, it reads like WGA was used to measure fibrosis.
  • Lines 272 beneficial spacing
  • Lines 275-277 is weak and very broad. Need to be more specific about lipids and proteins
  • Line 356 should read modulates

Author Response

Dear Editors and reviewers:
Please see the attachment.

Round 2

Reviewer 1 Report

ok, the concerns were added to the manuscript

the concerns were added to the manuscript

Author Response

Dear Editors and Reviewers

    Thank you for your letter and for the reviewers’ comments concerning our manuscript entitled “Aerobic exercise alleviates cardiac dysfunction correlated with lipidomics and mitochondrial quality control” (ID: 3589087).Those comments are all valuable and very helpful for revising and improving our paper, as well as the important guiding significance to our researches. 

Reviewer 2 Report

I have a few concerns that needs to be addressed.

Q4: In Figure 5, the authors figure heading reads, “Aerobic exercise restores lipidomic disturbances and cardiolipin acyl composition in left heart cardiomyocytes of heart failure mice”. However, none of the presented results show any sort of significance in the data to warrant this strong conclusion. The authors need to revisit this finding to make sure that the extrapolated results support their conclusion.

Response: We set the test period at a 10-week period in order to approximate the symptoms of heart failure patients. This longer trial period may have caused the lipidomic metrics in Figure 5 to miss the period of significant difference. However, our Figure 4 illustrates significant differences in the phospholipids that play a key role in the improvement of heart failure with exercise, such as cardiolipin.

New Query:

  • While I understand that fig 4 shows significant differences in cardiolipin, the purpose of fig 5 is to examine how 4-HNE, PA and PE are impacted in the model. This still does not show any significance in it. Yet the authors fig 5 title reads,

“Figure 5. Aerobic exercise restores lipidomic disturbances and cardiolipin acyl composition in left 214 heart cardiomyocytes of heart failure mice. (a-b) Exercise restored the levels of 4-HNE and 4-HNE 215 (341.82) and the levels of PA and PA (20:4PA) (c-d). (e-g) Aerobic exercise restored the cardiac PE 216 and PE (18:0 PE) and PE (22:6 PE) level.”

This is still not an accurate representation of the data observed and hence needs to be either removed or worded accordingly. In lines 208-209, the authors mention, “We found compensatory elevations in the hearts of heart failure mice by assaying PE content in the left heart (Figure 4, 5). Abnormal expression of PE and elevation of lipid peroxidation 4-HNE further imply that mitochondrial dysfunction in heart failure forms a vicious cycle with 211 lipid damage.”

PE is not elevated in any of the figures, it is decreased in fig 4. 4-HNE is not significantly altered in HF. These statements need to be further revised.

  • The claim that mitochondrial division is downregulated as mentioned in lines 258-259 is not accurate just based on some protein data and no activity assay or morphological support. Please reword it.

Minor Comment

The provided original images must be labelled properly and complete images for B-actin9loading control) are not available for the blots.

I have a few concerns that needs to be addressed.

Q4: In Figure 5, the authors figure heading reads, “Aerobic exercise restores lipidomic disturbances and cardiolipin acyl composition in left heart cardiomyocytes of heart failure mice”. However, none of the presented results show any sort of significance in the data to warrant this strong conclusion. The authors need to revisit this finding to make sure that the extrapolated results support their conclusion.

Response: We set the test period at a 10-week period in order to approximate the symptoms of heart failure patients. This longer trial period may have caused the lipidomic metrics in Figure 5 to miss the period of significant difference. However, our Figure 4 illustrates significant differences in the phospholipids that play a key role in the improvement of heart failure with exercise, such as cardiolipin.

New Query:

  • While I understand that fig 4 shows significant differences in cardiolipin, the purpose of fig 5 is to examine how 4-HNE, PA and PE are impacted in the model. This still does not show any significance in it. Yet the authors fig 5 title reads,

“Figure 5. Aerobic exercise restores lipidomic disturbances and cardiolipin acyl composition in left 214 heart cardiomyocytes of heart failure mice. (a-b) Exercise restored the levels of 4-HNE and 4-HNE 215 (341.82) and the levels of PA and PA (20:4PA) (c-d). (e-g) Aerobic exercise restored the cardiac PE 216 and PE (18:0 PE) and PE (22:6 PE) level.”

This is still not an accurate representation of the data observed and hence needs to be either removed or worded accordingly. In lines 208-209, the authors mention, “We found compensatory elevations in the hearts of heart failure mice by assaying PE content in the left heart (Figure 4, 5). Abnormal expression of PE and elevation of lipid peroxidation 4-HNE further imply that mitochondrial dysfunction in heart failure forms a vicious cycle with 211 lipid damage.”

PE is not elevated in any of the figures, it is decreased in fig 4. 4-HNE is not significantly altered in HF. These statements need to be further revised.

  • The claim that mitochondrial division is downregulated as mentioned in lines 258-259 is not accurate just based on some protein data and no activity assay or morphological support. Please reword it.

Minor Comment

The provided original images must be labelled properly and complete images for B-actin9loading control) are not available for the blots.

Author Response

Dear Editors and Reviewers:

    Thank you for your letter and for the reviewers’ comments concerning our manuscript entitled “Aerobic exercise alleviates cardiac dysfunction correlated with lipidomics and mitochondrial quality control” (ID: 3589087).Those comments are all valuable and very helpful for revising and improving our paper, as well as the important guiding significance to our researches. 

     Thank you for your suggestion. We have modified Figure 5 in lines 214 to 219 of the manuscript. Regarding phospholipid changes in heart failure, we have modified lines 208 to 214 of the manuscript. We have rewrote the description of the results in lines 260 to 263 of the manuscript.Regarding the images of the original strips, we have included pictures of the original data in the attachment to the email dated 30 May.

Round 3

Reviewer 2 Report

The authors have done a commendable job to address my concerns.

The authors have done a commendable job to address my concerns.